# Reactor Temperature Control Based on Improved Fractional Order Self-Anti-Disturbance

**Xiaowei Tang** [1] **, Bing Xu** [1,*] **and Zichen Xu** [2]

1 School of Electrical and Electronic Engineering, Shanghai Institute of Technology, Shanghai 201418, China
2 Engineering Management, Melbourne University, Melbourne 3128, Australia
* Correspondence: xubing2060@163.com

**Abstract:** In the chemical industry, a reactor is an absolutely necessary container. The fact that its dynamic qualities are nonlinear and unknown, however, is what causes the temperature to deviate from the value that was specified. As a result, the typical PID control cannot fulfill the prerequisites of the production process. A new nonlinear function is presented to replace the function that was previously used, and a temperature controller that is based on better fractional order active disturbance rejection is devised. On the basis of a new fractional order temperature detector (FOTD), a new fractional order equilibrium state observer (FOESO), and nonlinear function, an improved fractional order active disturbance rejection controller has been developed. A model of the reactor was created, and the dynamic properties of temperature control were investigated. By simulation and experimentation, it was demonstrated that the strategy has a number of benefits and is effective. In this approach, the information provided by the model is exploited to its maximum potential, and the temperature of the inlet cooling water is employed as the temperature control disturbance for feedforward compensation. Over the entirety of the process, this guarantees that the desired temperature will be preserved. When compared to FADRC, PID, and ADRC, the rising time is increased by 5 s, and the overshoot is raised by 25%. It has been established that the fraction-order active disturbance rejection controller has a quicker response speed, a higher capacity for anti-interference, and a quicker speed of stabilization.

**Keywords:** fractional order; self-disturbance control; reactor; temperature control

## 1. Introduction

It is common knowledge that the production reactions in a reactor are complicated and nonlinear [1], which makes it difficult to establish accurate mathematical models. Most enterprises still stay in the traditional PID control, but its control cannot meet the needs of this strong nonlinear system in today's chemical enterprises. The question and challenge that has emerged as most important for companies that deal in chemicals over the past few years is how to ensure that the temperature is controlled within the controllable range while the reactor is operating under stable conditions. This question is closely related to the question of how to improve the product's quality. In recent years, a number of academics have argued that a self-anti-disturbance control technique is often used in industrial process control [2,3]. This idea has gained a lot of traction in the scientific community.

Wu et al. [4] proposed an enhanced ADRC-based cascade steam temperature control strategy, and the field application in a 300 MW power plant demonstrated the strategy's benefits by demonstrating that the temperature deviation can be significantly reduced under both small-scale and large-scale load variation conditions. Chen et al. [5] suggested a low-order self-adjoint control (ADRC) approach based on a phase compensation (PC) technique, and analysis and simulation results demonstrated that the PC may significantly enhance the resilience and rapid response of a higher-order process control system. Jin et al. [6] designed a temperature self-rejecting controller based on the dynamic mathematical model of the intermittent polymerization reaction process, and the results showed

that the self-rejecting controller can effectively realize the temperature control of the PTFE semi-intermittent polymerization reaction process, and can adapt to the external noise disturbance interference and different batch production requirements, with the advantages of high control accuracy and high control stability. Chen et al. [7] developed a fractional order self-rejecting controller with FO extended state fractional order self-adjoint control with an observer design and empirically demonstrated that the FO-ADRC control system is resilient to uncertain system dynamics and disturbances. Li et al. [8] introduced a fractional order active disturbance suppression control (FADRC) scheme, and numerical simulation results demonstrated that it is superior and more effective than existing ADRC solutions. Yi et al. [9] introduced a time-lag fractional order active disturbance suppression control (TD-FADRC) method, which can effectively avoid oscillations in the model and has greatly enhanced tracking and disturbance response performance in comparison to the FPID control strategy. Zhen et al. [10] have connected proportional-integral-differential (PID) control and active disturbance rejection (ADRC) control and proposed an optimal active disturbance rejection controller based on proportional differential (PD) control law. Experiments have shown that dynamic and steady-state control performance can be improved. Wu et al. [11] proposed an ADRC based on a Smith predictor. Compared with a proportional integral controller and conventional ADRC, simulations and experiments verified that the proposed control strategy has good input interference suppression and measurement noise suppression capabilities. Compared with the recently developed control method without anti-integral saturation compensation, the strategy applied to temperature control of jacket crystallizer is superior. In addition, some scholars have proposed other control methods about fractional order [12].

The above scholars have carried out the combination of research and application of the self-disturbance controller and have achieved good control effects, so based on the above theoretical research basis, this paper takes the reactor temperature control as the research object from the practical application of the pharmaceutical production condensation reactor. It combines the emerging algorithms from recent research on industrial control and proposes to combine the feedforward self-anti-disturbance control with the fractional order theory. Additionally, it incorporates the design concepts of over-arranging the process, expanding the state observer, and estimating compensation.

The following is an outline of the remaining parts of this work. In the second section, field measurements from the reactor are used to inform the construction of a mathematical model of the temperature inside the reactor. The approach for integrating feedforward self immunity with fractional order controller is discussed in Section 3, along with the implementation of the accompanying. The comparison of the simulation results of FADRC and ADRC is presented in the Section 4. This section also shows how the proposed control approach was applied to the process of regulating the temperature of the reactor. The report's findings and interpretations are presented in Section 5.

## 2. Reactor System Modeling

*Reaction Tank Modeling*

The structure and location of the enterprise reaction kettle are depicted in Figure 1 and Figure 2, respectively. It can be deduced from these figures that the reactor is made up of three essential parts: the reaction vessel, the stirring mechanism, and the jacket. The material can be fed into the vessel either by being pulled in through the solid feeder or by the vessel's intake. The equipment for stirring is made up of various parts, including a stirrer and a motor specifically designed for stirring. The jacket's primary function is to maintain a constant temperature within the reaction vessel, which is completely sealed off from the outside world by the vessel's rim. When it is necessary to bring the temperature down, the cooling water is poured into the jacket, where it works to remove heat from the kettle.

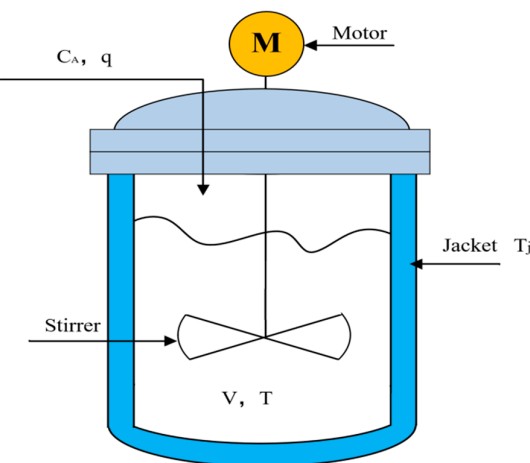

**Figure 1.** Reactive kettle structure.

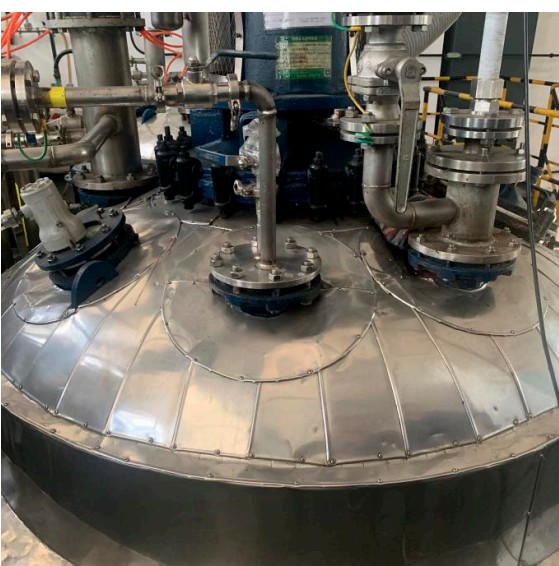

**Figure 2.** Site enterprise reaction kettle.

In order to develop a mathematical model of the reactor, the heat conservation that occurs within the kettle is utilized as a starting point. From this mathematical model, the process model of the reactor may be obtained by making an estimate of the reaction kinetics and energy balance equations. During a chemical reaction, the equation [13] developed by Arrhenius depicts the connection between the temperature on the inside of a reactor and the rate at which its internal reactants are reacting.

$$K = k_0 \exp\left(-\frac{E}{RT}\right) \tag{1}$$

When the temperature of the reactor is influenced by disturbance factors leading to a temperature rise, according to Equation (1) the reaction rate and exothermic rate in the kettle will also increase causing the reaction exotherm to accelerate. The temperature of the kettle rises, i.e., there is a positive feedback self-excitation relationship between the reaction temperature, and the reaction rate and exothermic rate [14], which, if not controlled, may result in "boiling." The occurrence of the phenomena known as "kettle explosion" may threaten the safety of production if it is not contained. In the early stage of the reaction, after the material has been added to the reactor, the reactant will not react chemically at room temperature and pressure to its full power heating to the required temperature of

the reaction. In this process, the reaction begins to gradually become exothermic. When the rate of temperature rise in the kettle is detected to exceed the rate of temperature rise limited by the process, it stops heating and relies on the reaction's exothermic energy to complete the reaction. When the rate of temperature rise in the kettle is detected to exceed the amount of heat that has accumulated in the kettle, it is equal to the amount of heat that has been transferred minus the amount of heat that has been lost during the process of transferring the heat. The reaction becomes quite exothermic as it reaches the zone where the temperature remains constant. The progression of the temperature during the process is depicted in Figure 3. The following is a description of the heat that is present in the reactor at this moment, as determined by the heat balance relationship [15]: heat in the reactor is equal to the sum of the heat created by the chemical reaction, the heat that is transferred, and the heat that is lost while the heat is being transferred.

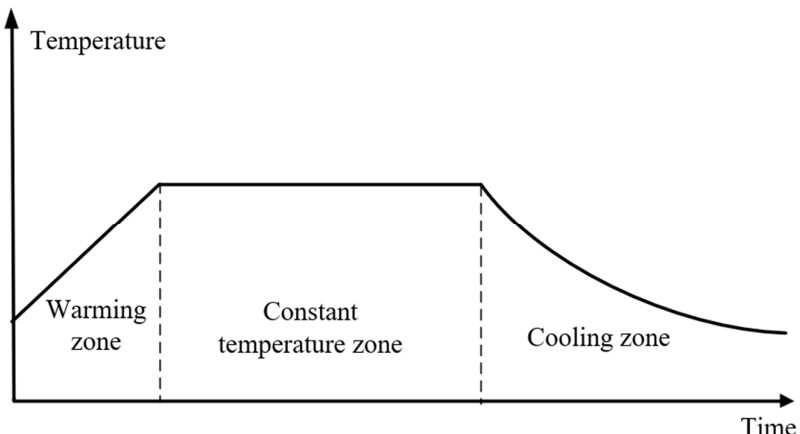

**Figure 3.** Temperature change curve.

The above relationship can be expressed visually as follows:

$$\frac{[\text{Heat accumulation in the kettle}]}{\text{Unit time}} = \frac{[\text{Heat from chemical reactions}]}{\text{Unit time}} \frac{[\text{Heat exchange}]}{\text{Unit time}}$$

The dynamic equilibrium equation for the heat in the reactor at this stage is shown in Equation (2)

$$vpc_p\frac{dT}{dt} = UA(T_C - T) - V(-\Delta H)r(C_A, T) \tag{2}$$

$$r(C_A, T) = -\frac{dC_A}{dt} = k_0 C_A e^{-\frac{E}{RT}} \tag{3}$$

Substituting Equation (3) into Equation (2) yields

$$\frac{dT}{dt} = \frac{UA}{v\rho c_p}(T_C - T) + \frac{(\Delta H)C_A}{\rho c_p}k_0 e^{-\frac{E}{RT}} \tag{4}$$

Collated from

$$\frac{dT_c}{dt} = \frac{F_c}{V_c}(T_j - T_c) + \frac{UA}{v_c\rho_c c_{pc}}(T_j - T_c) \tag{5}$$

To simplify the qualitative analysis of its dynamic behavior under perturbation using linear control theory, Equations (4) and (5) are linearized to obtain the matrix type of the linear equations, and the temperature model of the reactor is derived by simplifying this set of equations using differential incremental operations.

$$G(s) = \frac{T(s)}{F_C(s)} = \frac{a_{12}b_{22}}{s^2 - (a_{11} + a_{22})s + a_{11}a_{22} - a_{12}a_{21}} \tag{6}$$

The equations that came before it were constructed under slightly less realistic circumstances. Due to the fact that during the process of derivation the influence of the actual plant equipment as well as the other causes of the system are ignored, it is extremely difficult to attain the ideal state that was previously indicated when conducting production. The input for a closed-loop system is determined by taking the difference between the specified temperature value of the reactor and the actual measured temperature feedback value of the kettle. This allows for the creation of a system that is in a state of continuous feedback control. Table 1 shows the operating parameters of the reactor tank description, and Table 2 shows the operating parameters of the reactor tank.

**Table 1.** Operating parameters of the reactor tank description.

| Symbols | Description |
| --- | --- |
| $K$ | Reaction rate constant |
| $k_0$ | Reaction frequency factor |
| $E$ | Activation energy |
| $R$ | Molar gas constant |
| $T$ | Degree kelvin |
| $v$ | Reactant volume |
| $p$ | Reactant density |
| $c_p$ | Specific heat of reactant concentration |
| $C_A$ | Average concentration of reactants |
| $T$ | Kettle temperature |
| $A$ | Jacket heat transfer area |
| $U$ | Total heat transfer coefficient of jacket |
| $T_C$ | Jacket outlet temperature |
| $\Delta H$ | Molar heat of reaction |
| $T_j$ | Jacket inlet temperature |

**Table 2.** Operating parameters of the reactor tank.

| Process Variables | Parametric Values |
| --- | --- |
| Flow rate ($Q$) | 100 m$^3$/s |
| Volume ($V$) | 100 L |
| Jacket temperature ($T_{jt}$) | 280 K |
| Molar heat of reaction ($-\Delta H$) | 50,000 J/mol |
| Overall heat transfer coefficien ($UA$) | 200,000 Wb/K |
| Frequency factor ($k_0$) | $7.2 \times 10^{10}$ |
| Activation energy ($E$) | 9980 K |
| Mean concentration ($C_A$) | 0.08235 J/mol-K |
| Gas constant ($R$) | 8.3145 J/mol-K |
| Heat capacity ($C_p$) | 1 cal/gK |

## 3. Control Strategy of Reaction Tank

### 3.1. Fractional Order Controller

Since the introduction of the fractional order controller, it has seen widespread application across a variety of controller design applications [16–19]. As a result, the study of controllers has been expanded into a new discipline. It should come as no surprise that the fractional order controller's extension is the integer order controller. The traditional integer order controller has three adjustable parameters, but the fractional order controller has three adjustment parameters in total. The integer order controller has two more adjustment parameters than the fractional order controller. On the surface, it appears that the complexity of the calculations has increased; nevertheless, in reality, the controller's design notion has become more sensitive. The fractional order controller possesses excellent adaptability to parameter changes, along with flexible parameter modification, improved stability, and enhanced anti-interference capability [20–22]. Figure 4 depicts the FOPID control structure diagram in its entirety.

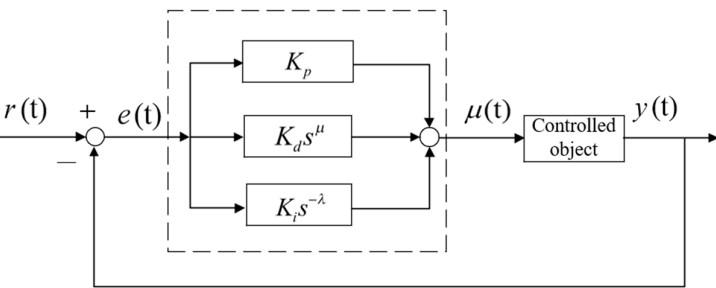

**Figure 4.** FOPID control structure.

The integral part affects the system's stability and dynamic performance, while the differential part brings the system closer to meeting the index requirements. From the response analysis, it is possible to conclude that fractional order PID control is superior to traditional PID control [23–26]. It has a mathematical model.

$$u(t) = K_p e(t) + K_i D^{-\lambda} e(t) + K_d D^{\mu} e(t)\mu \tag{7}$$

$$C(s) = K_p + K_i s^{-\lambda} + K_d s^{\mu}, (\lambda, \mu > 0) \tag{8}$$

Depending on the given values of $\lambda$ and $\mu$, the fractional order controller takes many forms as shown in Table 3.

**Table 3.** Operating parameters of the reactor tank.

| Value | Controller | Form |
|---|---|---|
| $\lambda = 0, \mu = 0$ | P | $C'(s) = K_p$ |
| $\lambda = 1, \mu = 0$ | IOPI | $C'(s) = K_p + K_i S^{-1}$ |
| $\lambda > 0, \lambda \neq 1;$ | FOPI | $C'(s) = K_p + K_i S^{-\lambda}$ |
| $\lambda > 0, \mu = 0$ | FO[PI] | $C'(s) = (K_p + K_i S)^{-\lambda}$ |
| $\lambda = 0, \mu = 1$ | IOPD | $C'(s) = K_p + K_d S$ |
| $\lambda = 0, \mu > 0,$ | FOPD | $C(s) = K_p + K_d S^{\mu}$ |
| $\mu \neq 1$ | FO[PD] | $C(s) = (K_p + K_d S)^{\mu}$ |
| $\lambda = 1, \mu = 1$ | IOPID | $C'(s) = K_p + K_i S^{-1} + K_d S$ |
| $\lambda, \mu > 0, \mu \neq 1;$ | FOPID | $C'(s) = K_p + K_i S^{-\lambda} + K_d S^{\mu}$ |

$\lambda$ and $\mu$ can be non-integers because the engineering field uses diverse performance indicators for the control effect, but the complicated form has not yet been utilized. For numerous engineering investigations, a proper adjustment of these two parameters is of great significance. According to the above explanation, the fractional order controller has two more adjustment parameters than the integer order controller, which appears to increase computing complexity, but in reality improves the controller's stability and resilience. It also reflects the control precision of the fractional order controller, which is more suited for engineering applications involving complexity.

### 3.2. Feedforward Self-Anti-Disturbance Controller Design

Controlling the flow of cooling water into the jacket regulates the temperature of the reactor. As the temperature control is an object with a large time constant, the control technique based on feedback error is slower to regulate the temperature; thus, feedforward control, which has advance compensation of disturbance signal, is more important [27]. The principal control method of feedforward control is founded on the idea of invariance, which is achieved by measuring the disturbance of the controlled object and compensating for the disturbance so that the controlled item's dynamic properties remain constant [28]. The algorithm is distinguished by its ability to provide the control quantity directly at the onset of disturbance generation, as opposed to controlling after the deviation of the

controlled quantity disturbance, which provides a certain override control capability in comparison to the error-based algorithm.

The core idea of the active disturbance rejection control algorithm is to set the series integral structure as the standard type and set the outside error as the total disturbance, which includes the external disturbance and the internal uncertain change of the model. The active disturbance rejection controller is composed of a tracking differentiator (TD), an extended state observer (ESO), and nonlinear feedback, and its control structure is shown in Figure 5. An extended state observer is designed to estimate the total disturbance and eliminate it in the control algorithm. This algorithm not only retains the characteristics of the traditional control independent model but also can estimate and compensate for the interference, so it has good anti-interference ability. The complete active disturbance rejection algorithm also includes a tracking differentiator for scheduling transition process and differential estimation, an extended state observer for estimating system state and interference, and nonlinear state error feedback for calculating control variables. TD arranges the desired system transition process, avoids system overshoot, and brings the expected input differential signal; ESO is used to estimate the state variables and all disturbances of the system, which is the priority of ADRC. The NLSEF is the initial control signal of the system generated by a nonlinear combination of errors between the transition process and the state estimator.

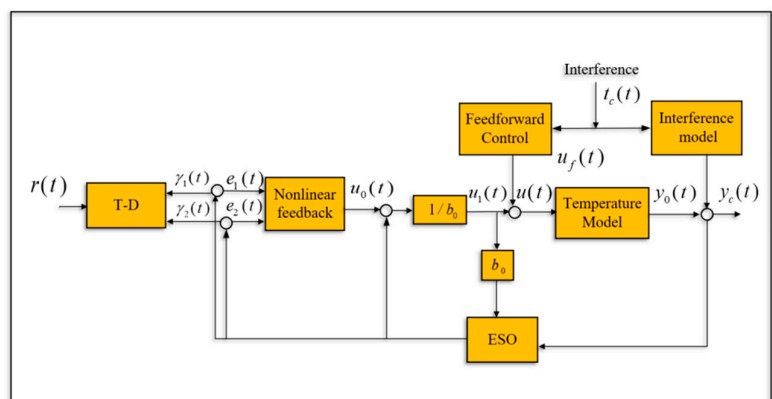

**Figure 5.** Feedforward self-anti-disturbance control structure.

*3.3. Fractional Order Feedforward Self-Anti-Disturbance Controller Design*

The self-anti-disturbance controller does not need to know the specific mathematical model of the controlled system and has a high degree of adaptability; it has the capability of automatic estimation to compensate for uncertain external disturbances; it solves the control problem of the unknown nature of the controlled object model and inaccurate parameters; and it considers the unknown external disturbance and the unmodeled system dynamics as the total disturbance of the system. In addition, the tracking differentiator in the self-anti-disturbance controller arranges the transition process to prevent the overshoot of the system The expansion state observer not only completes the estimation compensation of the controlled object disturbance error, but the nonlinear feedback law part also improves the anti-disturbance performance of the fractional order calculus controller. Due to the uniqueness of its structure, each control module does not interfere with each other, but the whole is connected, which means that the feedforward self-anti-disturbance control algorithm combined with other control algorithms will have a lot of research space.

Compared to traditional fractional order control, this method adds fractional order expansion state observer and fractional order tracking differentiator to integrate the internal and external disturbances of the system as unknown disturbances into the fractional order expansion state observer as the expansion state variables of the system to complete dynamic observation and compensation, which can significantly enhance the anti-disturbance performance of fractional order control. Figure 6 depicts the construction of the fractional

order feedforward self-anti-disturbance control system, revealing that FOADRC comprises three major components: FOESO, which is used to observe internal and external disturbances, and the system state. FOTD, which is prevent step response from causing severe overshoot.The FOPID controller is used to increase the system's control effect, where $r(t)$ represents the input signal and $y_c(t)$ represents the output signal of the system response.

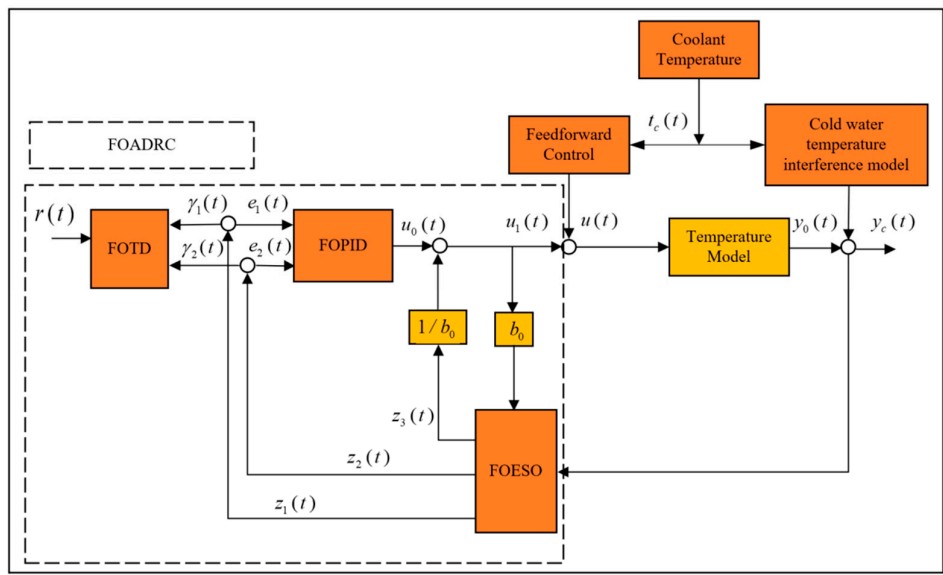

**Figure 6.** Fractional order feedforward self-turbulence control system.

### 3.3.1. Improving Nonlinear Functions

In ADRC, ESO and NLSEF are often in the form of nonlinear functions. The development and utilization of such nonlinear characteristics can bring great advantages to the control process; for example, it can be used to compensate the nonlinear in the controlled object, improve the control performance quality, and improve the robustness of the control system. The introduction of the function is based on the analysis of a large number of control examples.

When designing the nonlinear function, four aspects should be considered: (1) the function should have the characteristics of small error and large gain, and large error and small gain; (2) the function is continuous and differentiable everywhere; (3) the control quantity is not too large, which has a limited effect on output amplitude; and (4) the function should have easy parameter setting and simple function form.

Although it has the characteristics of large error and small gain, and small error and large gain, the $fal$ function-based controller is essentially a switching strategy between linear and nonlinear. The shortcoming is that the traditional nonlinear function at the origin and breakpoint is not differentiable, with a lack of satisfactory continuity and smoothness; therefore, based on the above four aspects, the b function is selected as the controller nonlinear function, and its expression is as follows:

$$atanh(x, b, k) = \frac{b^{kx} - b^{-kx}}{b^{kx} + b^{-kx}}(b > 1, k > 0) \tag{9}$$

### 3.3.2. FOTD

The tracking differentiator (TD) is primarily intended to pre-process the initial signal, to obtain the differentiated signal effectively by tracking the given signal faster, to add a transition process to the system, and to prevent the intense oscillation caused by the signal

jump, thus solving the problem that the system cannot simultaneously achieve fast and overshoot. The format of FOTD in this work is therefore as follows:

$$\left\{ \begin{array}{l} fh = fhan[x_1(k) - v(k), x_2(k), r, h_0] \\ x_1(k+1) = x_1(k) + hx_2(k) \\ x_2(k+1) = x_2(k) + h(n) \end{array} \right\} \tag{10}$$

$$\left\{ \begin{array}{l} d = rh_0 \\ d_0 = h_0 d \\ y = x_1 + h_0 x_2 \\ a_0 = \sqrt{d^2 + 8r|y|} \\ a = \left\{ \begin{array}{l} x_2 + \frac{(a_0 - d)}{2} sign(y), |y| > d_0 \\ x_2 + \frac{y}{h_0}, |y| \leq d_0 \end{array} \right. \\ fhan = - \left\{ \begin{array}{l} rsign(a), |a| \leq d \\ r\frac{a}{d}, |a| \leq d \end{array} \right. \end{array} \right. \tag{11}$$

### 3.3.3. FOESO

It will expand the perturbation of the output of the controlled object into a new state variable, treat all perturbations inside and outside the system as the total perturbation of the system, construct the state that can be observed to be expanded with a special feedback regime, estimate the state of the system and all perturbations in real-time, and provide some compensation. The expanded state observer is the most important component of self-disturbance control. It will expand the perturbation of the output of the controlled object into a new state variable. As a result, the FOESO is presented in the following manner in this study.

$$\left\{ \begin{array}{l} e_1 = y \\ \dot{z}_1 = z_2 - \beta_{01} \\ \dot{z}_2 = z_3 - \beta_{02} atanh(x, e_1, k_1) + b_0 u \\ \dot{z}_3 = -\beta_{03} atanh(x, e_1, k_2) \end{array} \right. \tag{12}$$

$$\left\{ u = u_0 - \frac{z_n}{b_0}, u_0 = \beta_1 atanh(x, e_1, k_1) + \beta_2 atanh(x, e_1, k_2) \right. \tag{13}$$

$\beta_{01}, \beta_{02}, \beta_{03}$ represent the observer parameter, $z_1, z_2, z_3$ represent the estimated output of the observer, output differentiation, and estimated total disturbance, $e$ represents the estimated error, and $\beta_1, \beta_2$ represent the observer's correction parameter; $b_0$ represent the compensation factor.

### 3.3.4. FOPID

According to the known transfer function Equation (6) of the reactor and the known transfer function Equation (8) of the fractional PID controller, in order to make the temperature of the control system in the frequency domain, the phase margin and amplitude margin are taken as the basis for the design of the fractional PID controller. In advance of the known performance indicators, Equations (6) and (7) can be converted into

$$G(j\omega) = \frac{K}{(j\omega)^2 + 2\rho\omega_n j\omega + \omega_n^2} e^{-\tau j\omega} \tag{14}$$

$$G_{jc}(j\omega) = k_p + k_i(j\omega)^{-\lambda} + k_d(j\omega)^{\mu} \tag{15}$$

where, $K$ is system gain, $\rho$ is damping ratio, and wn is natural frequency. Where the phase margin $\phi_m$, cut-off frequency $\omega_{cg}$, and amplitude margin $A_m$ should meet the following laws:

$$\begin{cases} \phi_m = \arg\left[G_{fc}(j\omega_{cg})G(j\omega_{cg})\right] + \pi \\ \left| G_{fc}(j\omega_{cg})G(j\omega_{cg})\right| = 0dB \\ \arg\left[G_{f_c}(j\omega_{cg})G(j\omega_{cg})\right] = -\pi \\ \left| G_{f_c}(j\omega_{cy})G(j\omega_{cg})\right| = \frac{1}{A_m} \end{cases} \qquad (16)$$

Set cut-off frequency, adoption period and phase margin as Wc = 10 rad/s, T = 0.4 s and $\phi_m$ = 70, respectively, and adopt the parameter setting rules of the controller such as phase margin, gain robustness, and nonlinear optimization function. The parameters of the fractional order PID controller can be obtained by calculation as follows: $k_p$ = 2.412, $k_i$ = 12.846, $k_d$ = 0.045, $\lambda$ = 0.743 and $\mu$ = 0.247. It is concluded that the transfer function of the fractional order PID controller is

$$G_c(s) = 2.412 + 12.846s^{-0.743} + 0.045s^{0.743} \qquad (17)$$

## 4. Simulation Experiments

In the experiment part, we use Matlab for simulation. A simulation model is built based on the process model that was established in Section 2, and the controller that was designed in Section 3 is used for simulation analysis and comparison with the classical self-rejecting controller, feedforward self-rejecting controller, and improved fractional order self-rejecting controller. This is done so that the efficacy of the proposed fractional order feedforward self-rejecting controller can be demonstrated. The integrated absolute error IAE of the response curve, the total variation TV of the control signal input, and rise time, overshoot are employed as comparison performance measurements.

The magnitude of the disturbance brought on by the introduction of the cooling water flow is illustrated in Figure 7, and it is plain to see that the disturbance is large. Figure 8 is a representation of the effect of feedforward control, and it can be observed that the system is not severely affected. This indicates that the feedforward control has effectively played a part in compensating for the perturbation and increasing the control quality of the system.

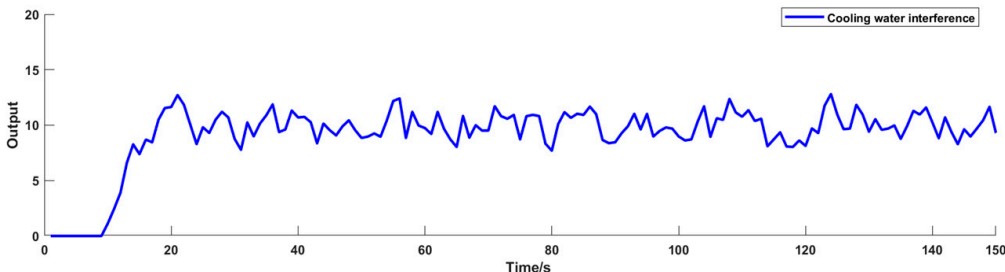

**Figure 7.** Cooling water flow disturbance.

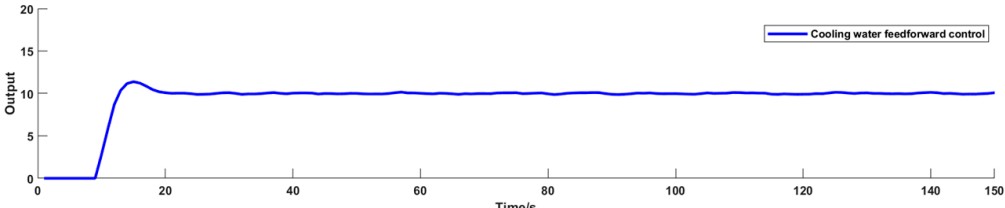

**Figure 8.** Feedforward control effect.

Figures 9 and 10 show the response curves and control signals for the four controllers, respectively. According to the findings of the comparison, the improved FOADRC has

a response time that is significantly less than that of the conventional ADRC, FADRC, and PID. Figure 9 demonstrates that the feedforward ADRC control provides superior anti-interference performance compared to the conventional ADRC and PID control.

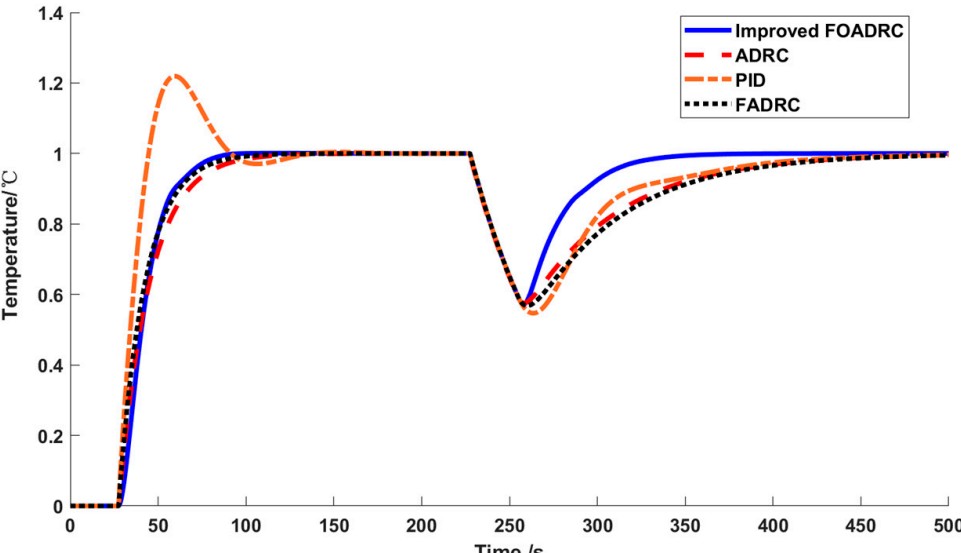

**Figure 9.** Response curve.

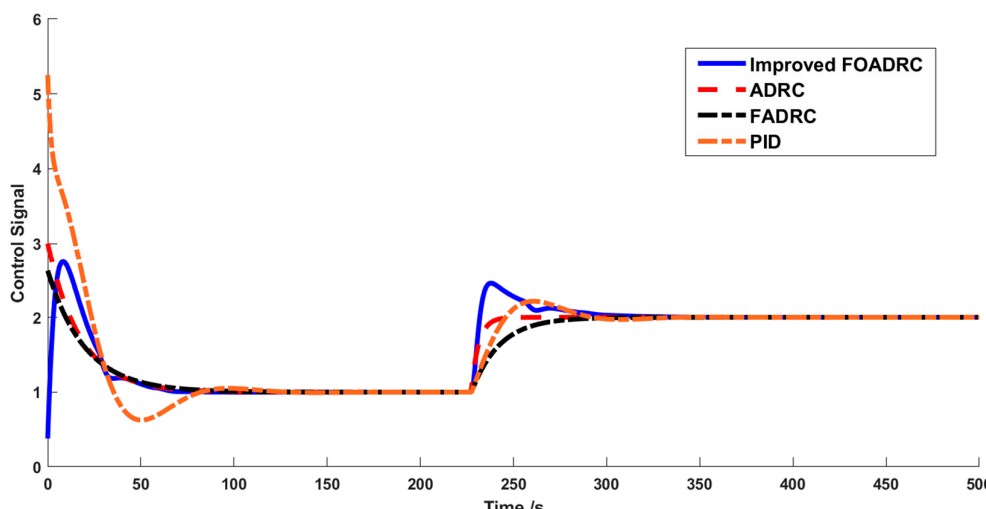

**Figure 10.** Control signals.

Figures 11 and 12 depict the response curves and control signals of the four controllers with the model modification when the real lag and time constant of the regulated process are considered to be 10% bigger than the model. Compared to the performance of the other three techniques, it is evident from the data that the improved FOADRC robustness has a considerable impact on control. Table 4 displays the performance indices of several control strategies, with $IAE_1$ and $TV_1$ being the performance indices following a 10% model modification. By comparing the performance indexes, the overshoot of the improved FOADRC controller is increased by 25%, and the rise time is increased by 5 s, which has a good control effect and verifies the effectiveness of the proposed method.

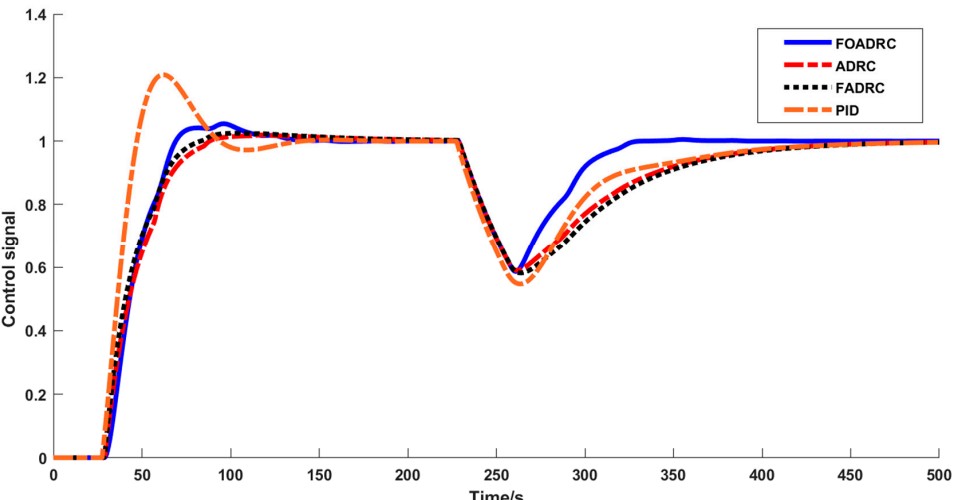

**Figure 11.** Model modification +10% response curve.

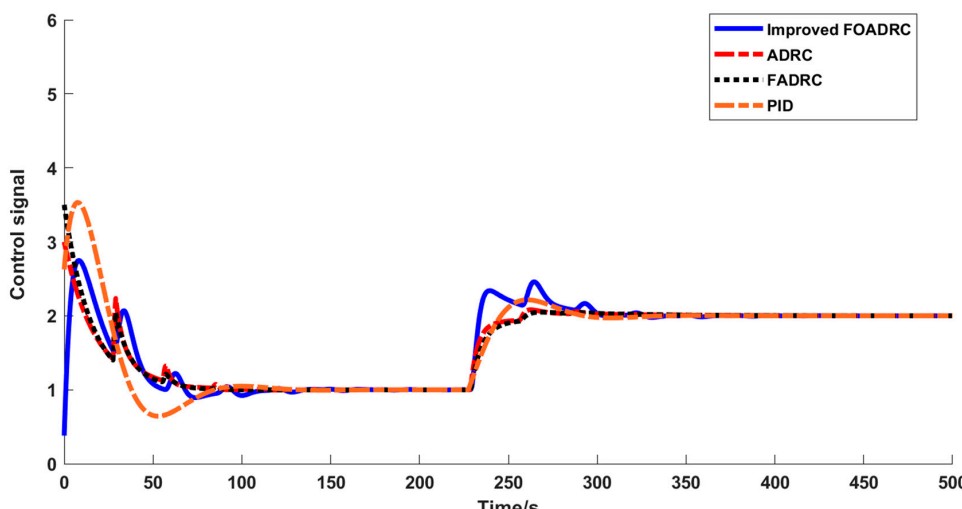

**Figure 12.** Model modification +10% control signals.

**Table 4.** Parameter Optimization for Different Algorithms.

| Control Strategy | IAE | IAE$_1$ | TV | TV$_1$ | Rise Time (s) | Overshoot (%) |
|---|---|---|---|---|---|---|
| Improved FOADRC | 42.75 | 43.74 | 1.00 | 1.12 | 35 | 2.2 |
| ADRC | 45.25 | 48.26 | 1.62 | 1.72 | 55 | 3.5 |
| FADRC | 43.75 | 46.56 | 1.12 | 1.30 | 52 | 5.2 |
| PID | 55.23 | 58.45 | 2.21 | 3.14 | 40 | 25 |

The simulation plots of the three sinusoidal signal processing methods for an input value of 1 are depicted in Figure 13. While working with a sinusoidal input signal having a value of 1, the figure illustrates that the enhanced FOADRC control strategy outperforms both the ADRC and FADRC control strategies in every way. This is demonstrated by the fact that the enhanced FOADRC control strategy has a higher value. The overshoot is decreased by 2%, the rise time is lowered by 0.7 s, and the tracking curve of FOADRC has less fluctuation and faster reaction than that of the usual ADRC. This is consistent with the curve of the input signal, which further validates the control performance of the FOADRC. The effectiveness of several strategies is compared in Table 5, which can be seen here.

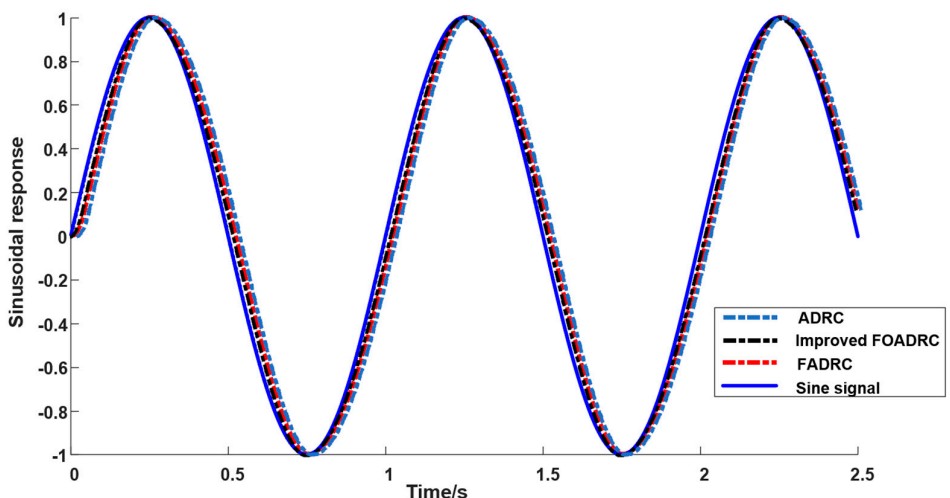

**Figure 13.** Sine signal response curve.

**Table 5.** Comparison of different control performance indicators.

| Control Strategy | Rise Time (s) | Overshoot (%) | Peak Time (s) |
|---|---|---|---|
| Improved FOADRC | 0.18 | 0.2 | 0.24 |
| ADRC | 0.35 | 2.3 | 0.35 |
| FADRC | 0.20 | 1.1 | 0.32 |

Sinusoidal signal tracking experiments of FADRC, ADRC and improved FADRC controller are compared. The performance index of the improved FOADRC controller is obviously better than that of the ADRC and FADRC controllers. In summary, the controller proposed in this paper has obvious advantages and verifies the effectiveness of the improved controller.

## 5. Conclusions

This work first conducts mathematical modeling on the reactor, and then presents an improved fractional order ADRC control strategy based on the reactor model. The goal of the modeling is to improve the performance of the temperature control system for the reactor. A fresh nonlinear function has been incorporated into the ADRC controller in order to facilitate its further development. An enhanced fractional active disturbance rejection (FOADRC) controller has been constructed. This controller is based on the new FOTD, the new FOESO, and a nonlinear feedback control law. In comparison to the conventional ADRC, PID, and FADRC numerical simulation results, the improved FOADRC strategy that was proposed can significantly improve the system performance and robustness, and it is also capable of successfully avoiding the interference that can occur in the process of controlling the temperature of the reactor. The response data demonstrate that the technique that was described in this study has greatly improved the response performance when compared with the ADRC, PID, and FADRC controllers. This improvement can be seen in both the tracking and jamming elements of the response. Not only is it applicable to the system that regulates the temperature of the reactor, but it may also be used for other high-order systems.

The proposed method for optimizing controller parameters has broad research prospects in the design of a variety of active disturbance rejection controllers because there are certain research needs to be addressed in the future regarding the optimization of controller parameters for fractional active disturbance rejection controllers.



**Author Contributions:** Conceptualization, X.T. and Z.X.; methodology, X.T.; software, X.T.; validation, X.T. and B.X.; formal analysis, X.T. and Z.X.; investigation, X.T.; resources, X.T.; data curation, X.T.; writing—original draft preparation, X.T.; writing—review and editing, Z.X. and B.X.; visualization, X.T.; supervision, B.X.; project administration, B.X. All authors have read and agreed to the published version of the manuscript.

**Funding:** This research received no external funding.

**Informed Consent Statement:** Not applicable.

**Data Availability Statement:** Not applicable.

**Conflicts of Interest:** The authors declare no conflict of interest.

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
