# Peer review of "Reactor Temperature Control Based on Improved Fractional Order Self-Anti-Disturbance"

_processes, doi:10.3390/pr11041125_

Round 1
Reviewer 1 Report
To start with, I would like to thank authors for their work in terms of interesting topic and well written article.
The paper is devoted to the design of improved fractional active disturbance rejection (FOADRC) controller of reactor temperature in the chemical industry. Within the controller developments paper proposes a new tracking differentiator and expanded state observer. Simulation results shoed the outperform of the developed FOADRC over ADRC and FADRC controllers.
Strengths:
· The theme of article is in the scope of SI topic “Advanced Performance-Oriented Evaluation, Diagnosis and Fault-Tolerant Control Techniques for Industrial Processes”.
· Abstract is adequate to article content
· Conclusion give the main findings
· References are sufficient ( but recent ones 2022 can be added), no self-citation is detected
· Material is almost cohesive and coherent
· Figures are clear (except of figure 7)
Drawback and my concerns:
1. Can the developed controller be used in the other industry applications? If yes, I recommend to set chemical reactor as a case study so as not to narrow application. For example, average fractional order controllers are also applied in motor speed controlling like in (https://doi.org/10.3390/en16041901). Please, Study it.
2. What are the outcomes of paragraph of literature review (line 40-71). The paragraph is not cohesive without deduction.
3. Give more information about chemical reactors. Where are they used, for what processes. In my opinion, it should be included.
4. Why does a chemical reactor need developing a new controller? What are used currently? What are they weak links in temperature control systems of reactors.
5. Line 23-24. “It has been established that the fraction-order active disturbance rejection control algorithm has a quicker response speed..” Where do you consider algorithms?
6. Discussion section is weak. We have a comparison of FOADRC with ADRC and FADRC controllers. What about other existing methods of reactor temperature controlling?
7. Control performance indicators was obtained for FOADRC, ADRC and FADRC controllers. The question is will the difference in these indicators be important in the process of temperature controlling? Maybe performance indicators of ADRC or FADRC controllers will be enough for it.
8. Feasibility study is not considered (optionally)
9. All formulas and symbols must be defined.
10. The paper needs proofreading and the revision of other small problems. Below you will find what I noticed :
Line 22 : (FADRC and ADRC) Define abbreviations and acronyms the first time they are used in the text
Line 38 : ( ..process control [2-3]. [2] This idea..) reference [2] appears two times
Line 120 : (the The) . Double “The”
Line 164-266: Just check
Author Response
We sincerely hope that this revised manuscript addresses all of your comments and suggestions. We thank the reviewers for their enthusiastic work and hope to be acknowledged. Thanks again for your comments and suggestions.

Reviewer 2 Report
This paper uses an improved fractional order active disturbance rejection controller for reactor temperature control. The proposed new method included a fractional order temperature detector (FOTD) and fractional order equilibrium state observer (FOESO). A simulation model of the dynamic properties of temperature control was also created.
My suggestions that may bring more clarity to this work are as follows.
1. A flowchart of the whole research algorithm would be useful for a better understanding of it.
2. The authors specified that the results were experimentally validated. How could the simulation model describe the experiment?
3. What are the specifications of the reaction kettle shown in Figure 2, including the main dimensions and working parameters?
4. A validation of the results against those presented in the literature would be welcome.
Author Response
We sincerely hope that this revised manuscript has addressed all your comments and suggestions. We appreciated for reviewers’ warm work earnestly,and hope that the correction will meet with approval.Once again,thank you very much for your comments and suggestions.
We would like to thank the referee again for taking the time to review our manuscript!

Reviewer 3 Report
The article deals with improvement of fractional-order control systems. The practical significance of the proposed approach is substantiated to need in enhancement of up-to-date approaches in reactor systems' modeling.
However, despite the article will be potentially interesting for an international scientific audience in process engineering, the following clarifications are needed:
1. It’s unclear what are the reason to consider fractional-order system (time-delay, anomalous behavior of the working environment, etc.). Please clarify what terms of equations (1)–(6) need to be considered as fractional order.
2. Due to the presented solutions (1)–(6) are well known, please substantiate the scientific novelty of the article more clear.
3. In the FOPID control system, fractional parameters “lambda” and “mu” can be changed only in a physically-substantiated range. What is this range? This particularly depends on the nature of actual forces in the reactor.
4. Due to the presented transfer function, the system can lose the dynamic stability. Please mention a couple of phrases about the stability of the considered system. Or maybe it’s a direction for further investigations?
5. What is the need in Fig. 7 from the scientific point of view? Because this is a well-known behavior of “atanh” function and its derivative. Such functions are widely used. However, what factors substantiate the use of a specific function?
6. In 3.3, equations (10)–(13) don’t consider fractional-order parameters. In this case, what is the need to use fractional-order PID systems?
7. How precise the results were obtained? What means did you use to evaluate the reliability of the proposed approach?
8. A deep discussion of the obtained results are needed before conclusions.
9. The conclusions are quite declarative. Please indicate the quantitative indicators that allows you substantiating the reasonability of using fractional-order control systems.
Author Response
We sincerely hope that this revised manuscript addresses all of your comments and suggestions. We thank the reviewers for their enthusiastic work and hope to be acknowledged. Thanks again for your comments and suggestions.
We would like to thank the referees again for taking the time to review our manuscripts!

Round 2
Reviewer 1 Report
Dear authors,
Thank you for the revision of work.
i have aready told you that the paper was good but requires addressing some concerns (which would make the paper better ). You have addressed many my concerns, but the number 7 is no, namely "Maybe performance indicators of ADRC or FADRC controllers will be enough for it." Is it worth to use the controller proposed by you in the the process of temperature controlling? I mean that the obtained performance indicators will give clear benefits ? or it is easier to use the traditional PID controller? Also, I do recommend indicating the reason of the developing the controller proposed in paper, in particular by adding up-to-date references (2022) what I recommend before
Author Response
Thank you again for your valuable comments on our paper.
We sincerely hope that this revised manuscript has addressed all your comments and suggestions. We appreciated for reviewers’ warm work earnestly,and hope that the correction will meet with approval.Once again,thank you very much for your comments and suggestions.

Reviewer 2 Report
The authors have answered all questions and taken all suggestions on board.
I recommend the publication of this work.
Author Response
Thank you again for your valuable comments on our paper.
Reviewer 3 Report
The authors have made all the efforts to improve the results' representation/clarification significantly. Therefore, the manuscript can be considered for publication.
Author Response

(The authors gave the same response as above.)

Round 3
Reviewer 1 Report
Dear authors, thank you for your explanation. I have no questions more